# Relationships between Internal Training Intensity and Well-Being Changes in Youth Football Players

**DOI:** 10.3390/healthcare10101814

**Published:** 2022-09-21

**Authors:** Rui Miguel Silva, Filipe Manuel Clemente, Francisco Tomás González-Fernández, Hadi Nobari, Rafael Oliveira, Ana Filipa Silva, José María Cancela-Carral

**Affiliations:** 1Faculty of Educational Sciences and Sports Sciences, University of Vigo, 36005 Pontevedra, Spain; 2Escola Superior Desporto e Lazer, Instituto Politécnico de Viana do Castelo, Rua Escola Industrial e Comercial de Nun’Álvares, 4900-347 Viana do Castelo, Portugal; 3Research Center in Sports Performance, Recreation, Innovation and Technology-SPRINT, 4900-347 Viana do Castelo, Portugal; 4Instituto de Telecomunicações, Delegação da Covilhã, 1049-001 Lisboa, Portugal; 5Department of Physical Education and Sport, Faculty of Education and Sport Sciences, Campus Melilla, University of Granada, 52006 Melilla, Spain; 6Department of Exercise Physiology, Faculty of Educational Sciences and Psychology, University of Mohaghegh Ardabili, Ardabil 56199-11367, Iran; 7Sepahan Football Club, 81887-78473 Isfahan, Iran; 8Department of Motor Performance, Faculty of Physical Education and Mountain Sports, Transilvania University of Braşov, 500068 Braşov, Romania; 9Faculty of Sport Sciences, University of Extremadura, 10003 Cáceres, Spain; 10Sports Science School of Rio Maior–Polytechnic Institute of Santarém, 2040-413 Rio Maior, Portugal; 11The Research Centre in Sports Sciences, Health Sciences and Human Development (CIDESD), 5001-801 Vila Real, Portugal; 12Life Quality Research Centre, 2040-413 Rio Maior, Portugal

**Keywords:** athletic performance, hooper index, load monitoring, perceived exertion, soccer, testing, wellness

## Abstract

The aim of this study was two-fold: (i) to analyze the variations of training intensity and well-being measures of youth football player over a 38 week period; and (ii) to test the relationships between training intensity and well-being variations throughout a youth football season. This study followed a longitudinal design, lasting 38 weeks. Twenty-five players (age: 15.0 ± 0.4 years) participated in this study. Participants were monitored daily to quantify the training intensity (TI) using the session-rate of perceived exertion (s-RPE) and its related indices: training monotony (TM), weekly training intensity (wTI), mean training intensity (mTI), and 5-day average (5d-AVG). A four-item questionnaire was collected daily to quantify the well-being status of each player. Fatigue, stress, delayed onset muscle soreness (DOMS), sleep quality, and the Hooper Index (HI) measures were analyzed. Kruskal-Wallis H test revealed significant differences for TM, mTI, wTI and 5d-AVG (H = 359.53, *p* = 0.001, η2 = 0.35; H = 414.97, *p* = 0.001, η2 = 0.41, H = 258.42, *p* = 0.001, η2 = 0.24 and H = 282.18, *p* = 0.001, η2 = 0.26). A positive large correlation was found between TM and sleep quality (r = 0.65, *p* = 0.05) and a negative large correlation between mTI and sleep quality (r = −0.69, *p* = 0.04). In conclusion, both TI and well-being measures present significant between-week differences at different periods of the season. Also, the variations of sleep quality seem to depend on TM and mTI variations across a youth football season.

## 1. Introduction

The implementation of training monitoring processes to follow players’ condition on a daily basis allows coaches and practitioners to better adjust training, supported by a more effective decision-making process [1]. Prescribing training while respecting training principles, such as individualization, progression and recovery is of paramount importance to ensure that positive adaptations occur [2]. Although not all teams have access to equipment for the quantification of locomotor demands, it is still possible to gather daily information regarding psychophysiological demands without any cost [3]. For clarity, throughout the present study, the term “training intensity” (TI) is used instead of “training load”, as suggested by a recent study that explains the relevance of the misuse of the latter term [4].

TI can be quantified through two dimensions: (i) external intensity (the mechanical stimulus imposed on players); and (ii) internal intensity (the psychobiological responses to the imposed mechanical stimulus) [5]. The different TI measures can be easily quantified through the use of the rate of perceived exertion (RPE), based on the Borg’s CR-10 scale [6,7]. RPE has previously been proved to be a reliable and valid measure of TI [8]. From the RPE values obtained after each training session, it is possible to calculate the session-RPE (s-RPE) through the multiplication of the RPE value and the duration of the training session in minutes [9]. Previous studies have reported significant correlations between the s-RPE method and important external and internal TI measures [8,10]. Namely, strong to very-strong correlations were reported between the s-RPE method and the percentage of maximal heart rate, the Bannister, Edwards and Lucia’s TRIMP methods, and maximal oxygen uptake [8]. 

Thus, from the s-RPE, the weekly TI variation (training monotony, TM) and its magnitude (training strain, TS), can be obtained, as previously proposed [11]. The TM measure is a 7-day mean of TI divided by its standard deviation. The daily use of TM measure allows coaches to identify the weekly variability of the imposed TI, where values above 2.0 arbitrary units (AU) represent lower TI variability within a microcycle, and values below 2.0 AU represent higher TI variability [11]. While, the training strain is the sum of the weekly training intensity, multiplied by training monotony [12]. This latter measure, allows to determine if the imposed TI has a higher or lower magnitude, in which values above 6000 AU are associated with greater risk of injury and bad overreaching [11].

Youth football players are subjected to different stressful contexts, such as school activities, pressure from the parents and other contextual factors beyond football, which can impact their training and match performance [13,14]. With this in mind, the implementation of well-being questionnaires allows coaches to further analyze how young athletes are coping with daily demands [15]. Well-being measures, such as the delay onset muscle soreness (DOMS), sleep quality, stress, fatigue and the Hooper Index (HI), are the most widely used in team sports contexts, as they previously show significant interactions among them [16]. Based on the athlete monitoring cycle proposed by Gabbett et al. [17], the well-being measures are associated with TI. As such, TI should be increased when high well-being values are combined with low TI. On the other hand, TI should be decreased when low well-being values are combined with high TI [17]. Moreover, it was revealed that low well-being values combined with high TI magnitudes (TS) and low variation (TM), can impair performance and increases injury occurrence [18].

A recent systematic review showed small-to-moderate associations between well-being measures and TI (based on s-RPE values) [19]. Indeed, a more recent study conducted on 27 under-17 football players [20], revealed that only small magnitude correlations were found between well-being and TI (based on RPE and s-RPE values) variations over 17 weeks. However, when considering some of the s-RPE related indices (TM and TS), moderate-to-large associations were found between well-being measures and TM and TS on youth football players [21]. Similarly, weekly TS and TM variations showed small-to-large associations with weekly variations of sleep quality, stress, DOMS, Fatigue and HI during a full youth football season [22].

To date, few studies have tested the relationships between TI indices such as TM and well-being variations over a full-season of youth football [21,22]. To the best of the authors’ knowledge, no studies have analyzed the variations of the sum of the intensity of all sessions and matches (weekly training intensity, wTI), the mean of the intensity of all sessions and matches (mean training intensity, mTI), and the average of the intensity of only five training sessions (5-day average, 5d-AVG). A greater understanding of the dependencies between different dimensions of TI and well-being variations throughout a youth football season. Moreover, analyzing such associations according to each period of the season may increase the chances to effectively manage players utilization during both training and competition. Also, better decision-making may be ensured by coaches, as well as more proximal communication with players’ parents, in an attempt to unravel some doubts that may arise regarding the well-being values.

For those reasons, the aims of the present study were two-fold: (i) to analyze the variations of different TI and well-being measures of youth football players according to each period of the season; and (ii) to analyze the relationships between the variations of TI and well-being measures during an entire youth football season.

## 2. Materials and Methods

### 2.1. Study Design 

A prospective cohort study design was used in this study.

### 2.2. Setting and Context

An under-15 youth elite football team was monitored daily over a period of 38 weeks. The season was divided as follows: from W1 to W6 (Pre-season), from W7 to W22 (Early-Season), from W23 to W38 (In-Season) and from W1 to W38 (Overall). The study started at 02/08/2021 and ended by 25/04/2022. Players trained five sessions and one match per week during the season, with two days off on Tuesdays and Fridays. Daily TI measures were collected (s-RPE, TM, mTI, wTI and 5d_AVG) as well as well-being measures (fatigue, stress, DOMS, sleep quality and hooper index). However, the s-RPE values were not considered for the analysis. A correlational research design was also used to examine the relationships between the variations of TI and well-being measures during the season.

### 2.3. Participants

Twenty-five youth football players (age: 15.0 ± 0.4 years old; height: 175 ± 0.6 cm; body mass: 62.1 ± 7.0 kg) from an under-15 team were monitored. From the twenty-five players, fourteen were defenders, five were midfielders and six were attackers. In order not to reduce the statistical power, the analysis between positions was not considered. The inclusion criteria were: (i) all players had to participate in at least 90% of training sessions throughout the season; and (ii) reported well-being ratings before all training sessions and matches. The goalkeepers were excluded from the sample. Before the study to be conducted, all players and their parents signed a written informed consent form. Also, before the start of the study, all the study procedures and their related risks and benefits were detailed to all players and their legal representatives. This study followed the ethical recommendations for the study in humans as suggested by the Declaration of Helsinki (updated version from 2013).

### 2.4. Training Intensity Quantification

Subjective TI measures were collected on a daily basis. Ten to thirty minutes after every training session, each player reported the perceived intensity on a scale from 1−10, where 1 means “very light activity” and 10 means “maximal exertion” [6]. Each player had to give the perceived value without the influence of other players. Given that, it was ensured that each player answered to the question “How intense was your session?” individually. Furthermore, the total duration of each training session was recorded. Then, the RPE value attributed by each player was multiplied by the total training session duration in minutes to obtain the session-rate of perceived exertion (s-RPE) [11]. From the s-RPE values, its related indices were calculated [12]: (i) weekly training intensity (wTI) (sum of the intensity of all sessions and match); (ii) mean training intensity (mTI) (mean of the intensity of all sessions and match); (iii) 5-day average (5d-AVG) (average of the intensity of only five sessions); and (iv) training monotony (TM) (mean of training intensity during the seven days of the week divided by the standard deviation).

### 2.5. Well-Being Quantification

To quantify the well-being status of each player, a self-reported questionnaire consisting of a 7-point scale was used on a daily basis, whereby a value of 1 is very, very low and a value of 7 is very, very high for fatigue, stress and DOMS measures, while 1 is very, very bad and 7 is very, very good for sleep quality [23], (Table 1). Approximately 30 min before each training session, each player rated their perception of well-being using a custom-designed questionnaire on a portable computer tablet. The questionnaire included the following well-being dimensions: (i) fatigue; (ii) stress; (iii) DOMS; and (iv) sleep quality. After the players had answered the questionnaire, the Hooper Index was obtained and used for analysis. This latter measure is the sum of the four question ratings.

### 2.6. Statistical Analysis

Descriptive statistics are represented as mean ± standard deviation (SD). Data were not normally distributed, and thus, non-parametric tests were used. As such, the Kruskal-Wallis H test was used to analyze the variations of TI measures (TM, mTI, wTI and 5d-AVG), as well as the variations of well-being measures (fatigue, stress, DOMS, sleep quality and hooper index) from W1 to W6 (Pre-season), from W7 to W22 (Early-Season), from W23 to W38 (In-Season) and from W1 to W38 (Overall). Effect sizes were indicated with partial eta squared. The interpretation of the effect sizes, regardless of the sign, was as follows: very small (0.01), small (0.20), medium (0.50), large (0.80), very large (1.20), huge (2.0) as initially suggested by Cohen [24], and expanded by Sawilowsky [25]. The level of significance was set at α ≤ 0.05. Posteriorly, a Pearson’s correlation coefficient r was used to examine the relationship between training intensity and well-being variation throughout the season. The magnitude of the correlations were interpreted as follows [26]: (0.0–0.1) trivial; (0.1–0.3) small; (0.3–0.5) moderate; (0.5–0.7) large; (0.7–0.9) very large; (0.90–1.00) nearly perfect. Data were analyzed using Statistica software (version 13.3; Statsoft, Inc., Tulsa, OK, USA).

## 3. Results

A Kruskal-Wallis H test was performed with training intensity measures from week 1 to week 6 (Pre-season) and revealed significant effects for TM (H = 19.15, *p* = 0.001, η2 = 0.09), mTI (H = 22.91, *p* = 0.001, η2 = 0.12), wTI (H = 17.54, *p* = 0.001, η2 = 0.08) and 5d-AVG (H = 20.27, *p* = 0.001, η2 = 0.10). From week 7 to week 22 (Early season) revealed significant effects for TM, mTI, wTI and 5d-AVG (H = 187.65, *p* = 0.001, η2 = 0.45; H = 116.58, *p* = 0.001, η2 = 0.26, H = 131.75, *p* = 0.001, η2 = 0.30 and H = 133.24, *p* = 0.001, η2 = 0.30), respectively. From week 23 to week 38 (In-season) revealed significant effects for TM, mTI, wTI and 5d-AVG (H = 182.65, *p* = 0.001, η2 = 0.43; H = 156.16, *p* = 0.001, η2 = 0.36, H = 100.05, *p* = 0.001, η2 = 0.2 and H = 93.07, *p* = 0.001, η2 = 0.20). When considering the overall season (from week 1 to week 38), it was revealed significant effects for TM, mTI, wTI and 5d-AVG (H = 359.53, *p* = 0.001, η2 = 0.35; H = 414.97, *p* = 0.001, η2 = 0.41, H = 258.42, *p* = 0.001, η2 = 0.24 and H = 282.18, *p* = 0.001, η2 = 0.26). (See Table 2 for more information).

Regarding the well-being measures, from week 1 to week 6 (Pre-season) did not reveal significant effects. From week 7 to week 22 (Early season) revealed significant effects for fatigue, hooper index, sleep quality, DOMS, and stress (H = 86.94, *p* = 0.001, η2 = 0.18; H = 92.69, *p* = 0.001, η2 = 0.20; H = 45.07, *p* = 0.001, η2 = 0.07, H = 75.59, *p* = 0.001, η2 = 0.15 and H = 79.24, *p* = 0.001, η2 = 0.16), respectively. From week 23 to week 38 (In-season), revealed significant effects for fatigue, hooper index, sleep quality, DOMS, and stress S (H = 57.17, *p* = 0.001, η2 = 0.11; H = 61.18, *p* = 0.001, η2 = 0.12; H = 40.24, *p* = 0.001, η2 = 0.07, H = 52.84, *p* = 0.001, η2 = 0.10 and H = 34.66, *p* = 0.001, η2 = 0.05). Finally, from week 1 to week 38 (overall season), revealed significant effects for fatigue, hooper index, sleep quality, DOMS, and stress (H = 166.58, *p* = 0.001, η2 = 0.14; H = 86.55, *p* = 0.001, η2 = 0.08; H = 115.17, *p* = 0.001, η2 = 0.08, H = 145.41, *p* = 0.001, η2 = 0.11 and H = 184.04, *p* = 0.001, η2 = 0.16), respectively. (See Table 3 and Figure 1 for more information).

Positive large correlation was found between TM and sleep quality (r = 0.65, *p* = 0.05) and a negative large correlation between mTI and sleep quality (r = −0.69, *p* = 0.04). Crucially, marginal values were found between wTI and sleep quality (r = −0.59, *p* = 0.08). No correlation was found in the case of 5-day average. (See Table 4 and Figure 2 for more information).

## 4. Discussion

The aims of the present study were to analyze the variations of TI and well-being measures throughout an entire youth football season and to test the relationships between the variations of TI and well-being measures. Significant differences between weeks for all TI and well-being measures were found. A significant positive large correlation was found between TM and sleep quality, while a negative large correlation was found between mTI and sleep quality.

Regarding the observed between-week differences of all TI measures (TM, mTI, wTI and 5d-AVG) in all periods of the season (pre-, early, and in-season), our findings are somewhat in concordance with previous studies that analyzed the variations of TM and TS measures [21,22,27]. Indeed, a very recent systematic review that summarized the main evidence about TI measures variations over the season in youth soccer players, revealed that the TM measure showed higher variability (0.8 to 3.3 AU) throughout the in-season period [28]. However, the authors of that study [28], affirmed that there is a lack of studies (only seven) analyzing TM variations on youth football players. Also, from those seven reported studies, none of them analyzed TM across a full-season. Still, the significant between-week differences found in each season period for TM values, are relatively expected. That is, during certain microcycles of the season higher training volumes are imposed (overload), which usually result in lower training variability (higher TM values). On the other hand, during tapering microcyles, lower TM values are expected [29].

As a consequence of TM differences, it was also expected that the mTI, wTI and 5d-AVG shows significant differences during the season. That is because those measures consider the mean of TI with and without the inclusion of a match (mTI and 5d-AVG, respectively), and the sum of the weekly TI (wTI). However, there is no reference for the observed differences of mTI, wTI and 5d-AVG measures, that makes possible to make comparisons with other studies. Further studies are needed to generalize such findings about the observed differences of mTI, wTI and 5d-AVG measures.

The magnitude of the between-week differences found for well-being measures at the early- and in-season-periods were similar to those reported in a recent study conducted on 27 under-17 football players [20]. Specifically, that study [20] revealed similar magnitudes for fatigue (*p* < 0.001; η2 = 0.147) and for DOMS (*p* = 0.001; η2 = 0.127) but not for sleep quality (*p* < 0.001; η2p = 0.140), which had a small magnitude compared to the trivial magnitude found in the present study. However, our study followed a different design, i.e., 38 weeks were analyzed, while the aforementioned study [20] analyzed only 17 weeks, corresponding to the pre- and early-season. Still, another study conducted on 21 under-16 football players showed higher weekly variations for fatigue, stress, sleep quality, DOMS and hooper index (coefficient of variation [CV] ranging from −32.7% to 174% CV) during the in-season period [22]. This is in partial agreement with the findings of the present study, as significant differences were found for all well-being measures in early- and in-season periods. Nevertheless, the higher values and variations of well-being measures during the competitive periods can be a consequence of the TI imposed by coaches on athletes, depending on the level of the opponents as well as other contextual factors [13].

Similar findings regarding the relationships between the variations of TI and well-being measures observed in the present study were reported in the study of Nobari et al. [22], when considering the relationship between TM and sleep quality measures. Indeed, in that study, a significant positive correlation was observed (r = 0.515; *p* < 0.001) between TM and sleep quality variations. In contrast, in the present study, an unclear positive moderate relationship was found between TM and stress variations, while in the Hadi et al. study [22], a significant negative large association with wStress (r = −0.426; *p* = 0.003) was reported. On the other hand, another study that tested such relationships, reported moderate positive correlations between TM and well-being variations (fatigue, stress, DOMS, sleep quality and hooper index [r = 0.32 to 0.46]) [21]. These discrepancies between our findings and the few available studies [21,22], can be attributed to the heterogeneity of the studies design. Although the new insight that greater between-week differences of TM values seem to be associated with greater differences of sleep quality can be promising, further studies are needed to make possible to generalize such findings.

Interestingly, when considering the 5d-AVG measure (average TI of the weekly five training sessions), no significant correlations were found with any well-being measures. Although speculative, this finding may suggest that match intensity has more influence on well-being measures variations. Indeed, form our findings, it seems that with lower mTI (mean of the intensity of all sessions and match), higher values of sleep quality (bad sleep quality) are perceived by players. This finding is in concordance with the findings of a very recent study that suggested that higher TI levels are associated with greater levels of readiness and sleep quality [20]. As sleep quality has an enormous and determinant impact on athletes’ performance [30], ensuring a cautious planning of the imposed weekly TI measures, namely, TM, and mTI measures, can potentially improve the players’ sleep quality and readiness to train [20].

This study presented some limitations, among which the small sample size—only one team was analyzed—is particularly noteworthy. The non-inclusion of objective TI measures such as heart rate-based measures and global-positioning system-based measures are another limitation. In future, the inclusion of these variables could give greater insights regarding the study findings. The lack of consideration and monitoring of important contextual factors that could have influenced the results of this study is another limitation. Additionally, the influence of match participation of each player was not considered individually. Future studies should therefore include larger sample sizes and objective TI measures and should consider the duration of individual match participation. 

## 5. Practical Implications

The findings of the present study indicate the importance for coaches to consider the management of important TI variations and correlate them with sleep quality variations over time. This could potentially benefit the management of their players’ health and readiness to train, as sleep quality is dependent on the values of the mean training intensity and the training variability (TM) throughout a microcycle. Coaches must ensure higher mean training intensity and high training variability (lower TM values) in order to promote improved sleep quality values. Also, using the mean of training intensity considering the preceding match (mTI) can be more informative regarding well-being changes compared to not considering the preceding match. Therefore, coaches would benefit from the inclusion of a match intensity metric from the same training week as the mean training intensity. However, this latter TI analysis must be done with caution, as further studies are needed to generalize such TI analyses. 

## 6. Conclusions

The present study examined variations of TI and well-being measures and their relationships across a full youth football season. The findings revealed that while all TI measures varied in all season periods, the well-being measures underwent significant weekly variations only after the pre-season period. Sleep quality seems to be influenced by TM and mTI variations throughout the season. Also, the inclusion of match intensity may be a determining factor influencing the observed variations of well-being measures. 

## Figures and Tables

**Figure 1 healthcare-10-01814-f001:**
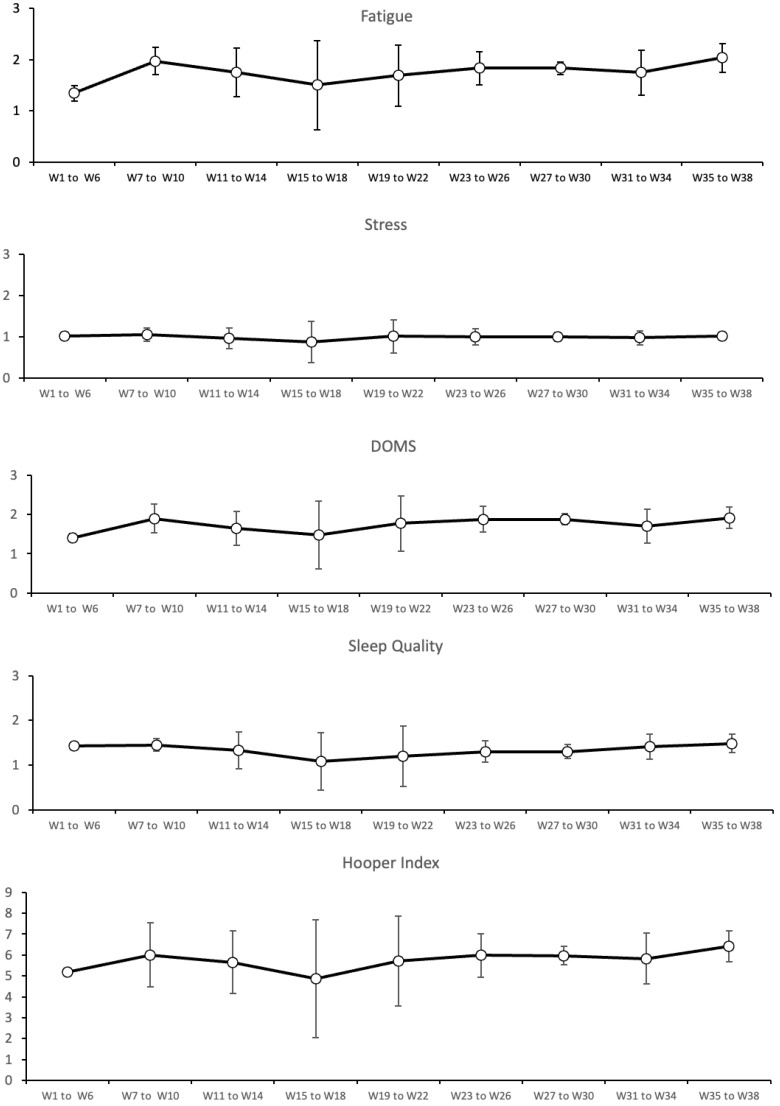
Variations of fatigue, sleep quality, delayed onset muscle soreness (DOMS), stress, and hooper index. W: weeks; A.U.: arbitrary units.

**Figure 2 healthcare-10-01814-f002:**
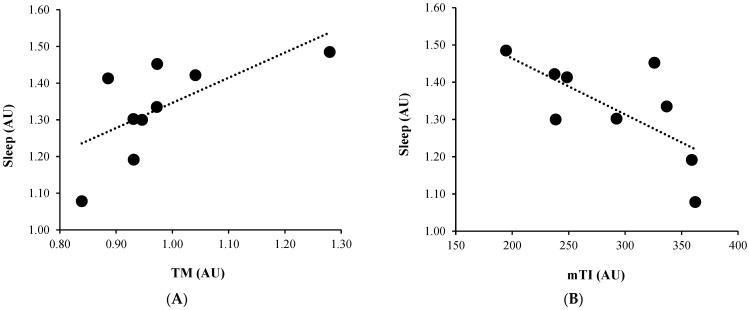
Significant correlations between TI and well-being measures. (**A**): Correlation between TM and sleep quality; (**B**): Correlation between mTI and sleep quality. TM: training monotony; mTI: mean training intensity; AU: arbitrary units.

**Table 1 healthcare-10-01814-t001:** Subjective ratings of fatigue, stress, delayed onset muscle soreness and quality of sleep.

Rating	Fatigue	Stress	DOMS	Sleep Quality
1	Very, very low	Very, very low	Very, very low	Very, very good
2	Very low	Very low	Very low	Very good
3	Low	Low	Low	Good
4	Average	Average	Average	Average
5	High	High	High	Bad
6	Very high	Very high	Very high	Very bad
7	Very, very high	Very, very high	Very, very high	Very, very bad

DOMS: delayed onset muscle soreness.

**Table 2 healthcare-10-01814-t002:** Training intensity variations from preseason, early-season, in season and overall.

	Pre-Season		Early-Season		In-Season		Overall Season	
	W1 to W6	H|*p*|η2	W7 to W22	H|*p*|η2	W23 to W38	H|*p*|η2	W1 to W38	H|*p*|η2
TI measures								
**TM (AU)**	0.99 ± 0.40	H = 19.15	0.92 ± 0.19	H = 187.65	1.00 ± 0.26	H = 182.65	0.97 ± 0.25	H = 359.53
		*p* = 0.001 **		*p* = 0.001 **		*p* = 0.001 **		*p* = 0.001 **
		η2 = 0.09		η2 = 0.45		η2 = 0.43		η2 = 0.35
**mTI (AU)**	235.77 ± 153.63	H = 22.91	367.30 ± 98.60	H = 116.58	294.18 ± 68.52	H = 156.16	315.75 ± 91.78	H = 414.97
		*p* = 0.001**		*p* = 0.001 **		*p* = 0.001 **		*p* = 0.001
		η2 = 0.12		η2 = 0.26		η2 = 0.36		η2 = 0.41
**wTI (AU)**	1123.25 ± 685.45	H = 17.54	1443.73 ± 478.89	H = 131.75	1337.02 ± 443.04	H = 100.05	1348.20 ± 496.41	H = 258.42
		*p* = 0.001 **		*p* = 0.001 **		*p* = 0.001 **		*p* = 0.001 **
		η2 = 0.08		η2 = 0.30		η2 = 0.22		η2 = 0.24
**5d-AVG (AU)**	231.68 ± 136.67	H = 20.27	292.23 ± 97.58	H = 133.24	243.05 ± 78.31	H = 93.07	261.96 ± 95.64	H = 282.18
		*p* = 0.001 **		*p* = 0.001 **		*p* = 0.001 **		*p* = 0.001 **
		η2 = 0.10		η2 = 0.30		η2 = 0.20		η2 = 0.26

W: weeks; TI: training intensity; TM: training monotony; TS: training strain; wTI: weekly training intensity; mTI: mean training intensity; 5d-AVG: 5-day average; AU: arbitrary units. ** denotes significance at *p* < 0.001.

**Table 3 healthcare-10-01814-t003:** Well-being variations from preseason, early-season, in season and overall.

	Pre-Season		Early-Season		In-Season		Overall Season	
	W1 to W6	H|*p*|η2	W7 to W22	H|*p*|η2	W23 to W38	H|*p*|η2	W1 to W38	H|*p*|η2
**Fatigue (AU)**	2.60 ± 07.13	H = 8.41	1.73 ± 0.69	H = 86.94	1.87 ± 0.67	H = 57.17	1.32 ± 0.45	H = 166.58
		*p* = 0.13		*p* = 0.001 **		*p* = 0.001 **		*p* = 0.001 **
		η2 = 0.02		η2 = 0.18		η2 = 0.11		η2 = 0.14
**Stress (AU)**	0.98 ± 0.59	H = 0.87	0.97 ± 0.39	H = 79.20	1.00 ± 0.34	H = 34.66	1.00 ± 0.30	H = 184.04
		*p* = 0.97		*p* = 0.001 **		*p* = 0.001 **		*p* = 0.001 **
		η2 = 0.02		η2 = 0.16		η2 = 0.05		η2 = 0.16
**DOMS (AU)**	1.38 ± 0.87	H = 5.74	1.65 ± 0.69	H = 75.59	1.84 ± 0.67	H = 52.84	1.30 ± 0.45	H = 145.41
		*p* = 0.32		*p* = 0.001 **		*p* = 0.001 **		*p* = 0.001 **
		η2 = 0.005		η2 = 0.15		η2 = 0.10		η2 = 0.11
**Sleep Quality (AU)**	1.39 ± 0.82	H = 0.30	1.30 ± 0.59	H = 45.07	1.35 ± 0.57	H = 40.24	1.16 ± 0.40	H = 115.17
		*p* = 1.00		*p* = 0.001 **		*p* = 0.001 **		*p* = 0.001 **
		η2 = 0.03		η2 = 0.07		η2 = 0.07		η2 = 0.08
**Hooper Index (AU)**	5.04 ± 3.12	H = 2.38	5.53 ± 2.00	H = 92.69	6.04 ± 1.88	H = 61.18	3.04 ± 1.13	H = 86.55
		*p* = 0.79		*p* = 0.001 **		*p* = 0.001 **		*p* = 0.001 **
		η2 = 0.01		η2 = 0.20		η2 = 0.12		η2 = 0.05

DOMS: delayed onset muscle soreness; AU: arbitrary units. ** denotes significance at *p* < 0.001.

**Table 4 healthcare-10-01814-t004:** Correlation between training intensity and well-being variations.

	Well-Being Measures				
TI Measures	Fatigue	Stress	DOMS	Sleep Quality	Hooper Index
TM	r = 0.41|*p* = 0.27	r = 0.47|*p* = 0.19	r = 0.31|*p* = 0.41	r = 0.65|*p* = 0.05 *	r = 0.56|*p* = 0.11
mTI	r = −0.22|*p* = 0.55	r = −0.38|*p* = 0.30	r = −0.22|*p* = 0.56	r = −0.69|*p* = 0.04 *	r = −0.48|*p* = 0.18
wTI	r = −0.17|*p* = 0.65	r = −0.30|*p* = 0.42	r = −0.18|*p* = 0.63	r = −0.59|*p* = 0.08	r = −0.43|*p* = 0.23
5d-AVG	r = −0.15|*p* = 0.68	r = −0.04|*p* = 0.90	r = −0.08|*p* = 0.83	r = −0.47|*p* = 0.19	r = −0.31|*p* = 0.40

TI: training intensity; TM: training monotony; mTI: mean training intensity; wTI: weekly training intensity; 5d-AVG: 5-day-average; DOMS: delayed onset muscle soreness. * Denotes significance at *p* ≤ 0.05.

## Data Availability

The datasets generated during and analyzed during the current study are available from the aim author or the corresponding author on reasonable request.

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
