# Peer review of "Relationships between Internal Training Intensity and Well-Being Changes in Youth Football Players"

_healthcare, 2022, doi:10.3390/healthcare10101814_

Round 1

Reviewer 1 Report

Dear authors,

The aim of this study was two-fold: (i) analyze the variations of training intensity and well-being measures of youth football player across weeks; and (ii) to test the relationships between training intensity and well-being variations throughout a youth football season.

It is an interesting manuscript with a research hot topic which can be accepted after minor revisions.

Please, consider the the attached revisions.

Kind regards.

Reviewer 2 Report

Dear authors, congratulations for this contribution. It is a quality work that addresses a topic of interest and that can be the basis for future research.

As the authors pointed out, it would be interesting for future research to use data on stress or rest from devices such as "Embrace" (empathic), "GoBe" (Healbe) or "Feel Emotion Sensor" (Sentio Solutions).

The study has a double purpose: (i) analyze the variations of training intensity and 28 well-being measures of youth football player across 38 weeks; and (ii) to test the relationships be-29 tween training intensity and well-being variations throughout a youth football season.

The topic is original and relevant in the field because prescribing training while respecting training principles, such as individualization, progression and recovery are of paramount importance to ensure that positive adaptations occur. The search and validation of methods that facilitate this task is of great interest and practical application.

This study revealed that while all training intensity measures vary between-week in all season-periods, the well-being measures have  significant weekly variations only after the pre-season period. The sleep quality seems to be influenced by the training monotony and mean training intensity variations across the season. Also, the inclusion of match intensity can be a determining factor influencing the observed variations of well-being measures.

The conclusions are consistent with the evidence and arguments presented and address the question posed.

The references are appropriate.

Reviewer 3 Report

I have no major concerns with this paper.  The paper is well written and the research design and methodology are appropriate.  I have no suggestions for changes or additions.  However, like many research papers on data collection from specific populations, I would like the authors to make a stance about what they have found.  They state that they found significant between-week differences at different periods of the season.  And, the variations in sleep quality seem to depend on TM and mT1 variations... Okay... so what do you recommend?  What is the meaningfulness of these findings?  Knowing that sleep quality is linked to performance, what should coaches do?  What about match intensity?  Should coaches be concerned with too often match intensity? I recommend that the authors should interpret their findings and offer some guidance for coaches.  Stating that coaches will be informed from their data, means little.  Coaches don't have the background to read this paper and apply it, that's why the authors need to give some meaning to their results.   If their study has meaning, and I think it does, then help coaches, inform them, and offer something useful to them. 
